# Effects of Thermomechanical Treatment on Phase Transformation of the Ti$_{50}$Ni$_{49}$W$_1$ Shape Memory Alloy

**Tyau-Song Huang [1], Shih-Fu Ou [1,*], Cheng-Hsien Kuo [1,*] and Cheng-Hsiu Yang [2]**

[1] Department of Mold and Die Engineering, National Kaohsiung University of Science and Technology, Kaohsiung 807, Taiwan; htsong@nkust.edu.tw

[2] Department of Materials Science, Feng Chia University, Taichung 407, Taiwan; chyang@fcu.edu.tw

\* Correspondence: m9203510@gmail.com (S.-F.O.); chuckkuo@nkust.edu.tw (C.-H.K.);
Tel.: +886-7-381-4526 (S.-F.O. & C.-H.K.); Fax: +886-7-383-5015 (S.-F.O. & C.-H.K.)

**Abstract:** The Ti$_{50}$Ni$_{49}$W$_1$ alloy has a B2 $\leftrightarrow$ 19' martensitic transformation but slightly lower shape recovery than the Ti$_{50}$Ni$_{50}$ alloy. The B19' martensite structure in the Ti$_{50}$Ni$_{49}$W$_1$ has the lattice parameters $a$ = 0.301 nm; $b$ = 0.423 nm; $c$ = 0.472 nm; and $\beta$ = 97.5°. The hardness increment and transformation temperature depression of Ti$_{50}$Ni$_{49}$W$_1$ are greater than those of Ti$_{50}$Ni$_{50}$ under the same degree of cold rolling and the same number of thermal cycles; owing to the Ti$_{50}$Ni$_{49}$W$_1$; with higher inherent hardness from solidification strengthening of W atoms. Both thermal cycling and cold rolling on Ti$_{50}$Ni$_{49}$W$_1$ also promotes R-phase transformation. The effects of thermal cycling and cold rolling on the martensitic transformation temperature ($M_s$) of the Ti$_{50}$Ni$_{49}$W$_1$ alloy follow a linear trend; and the $M_s$ decreased with the hardness.

**Keywords:** shape memory alloy; martensitic transformation; thermomechanical treatments

## 1. Introduction

TiNi-based alloys are among the most important shape memory alloys (SMAs) due to their significant shape memory effect and pseudoelasticity. Although TiNi alloys were discovered over 50 years ago, new applications are still being discovered for the biomedical industry, consumer products, and aerospace industry [1].

The thermoelasticity of SMAs relies on the B2 $\leftrightarrow$ 19' martensitic transformation behaviors, which can be influenced by heat treatments and plastic deformations, such as thermal cycling [2], aging treatment [3,4], and cold rolling [5]. Moreover, the martensitic transformation of TiNi SMAs is also affected by the addition of other elements. For instance, the martensitic transformation temperature ($M_s$) decreases when Ni is substituted by Cr, V, Mn, Fe, and Co [6–9], while the temperature decreases when amounts greater than 15–20 at.% of Au, Pt, and Pd are added to replace Ni [10–14].

The transformation sequence of TiNi-based SMAs can be B2 → R → B19' for the TiNiFe, TiNiAl, and TiNiCo alloys [9,15,16], or B2 → B19 → B19' for TiNiPd and TiNiCu alloys [17,18] in different thermomechanical processes involving a high-temperature cubic B2 phase, an intermediate rhombohedral R phase (or orthorhombic B19 phase), and a low-temperature monoclinic B19' phase. The addition of ternary elements can also provide strengthening effects to the TiNi SMAs. For instance, the matrix of TiNi alloy can be strengthened by solid-solution and second-phases precipitation by the addition of Cr with amounts less than 2 at.% [6,19]. Additionally, the shape recovery abilities of the Ti$_{51}$Ni$_{48.5}$Cr$_{0.5}$ and Ti$_{51}$Ni$_{49}$ alloys were improved to 96.7% and 88%, respectively [20]. Similar strength behavior also occurs in TiNiMo SMAs, which also have good corrosion resistance [21].

In this study, W was chosen to replace Ni in TiNi SMAs, since W has an atomic radius (0.137 nm) closer to Ni (0.125 nm) than to Ti (0.147 nm). In addition, W has a closer electronegativity (2.36) to

Ni (1.91) in comparison with the electronegativity of Ti (1.54) [22]. From the above information, it is expected that W atoms should substitute Ni atomic sites in the TiNi matrix. The common commercial application of SMAs was as shape-memory couplings for aircraft hydraulic connections. TiNiW SMAs were designed for increasing wear resistance of the TiNi SMAs by solidification strengthening.

The phase-transformation characteristics of TiNiW SMAs and the influences of thermomechanical treatments are rarely studied. In this study, we investigated the transformation behavior of the $Ti_{50}Ni_{49}W_1$ alloy. The effects of thermal cycling and cold rolling on the martensitic transformation of the alloy are also discussed.

## 2. Experimental Procedure

A $Ti_{50}Ni_{49}W_1$ ingot (about 120 g) was prepared, using a vacuum arc re-melter to melt and re-melt a Ti ingot (99.7 wt.%), Ni ingot (99.9 wt.%), and W ingot at least six times in an argon atmosphere. Homogenization comprised heating the as-melted ingots at 950 °C for 72 h, and then water quenching, followed by annealing at 1173 K for 2 h.

In this study, two procedures were conducted on the alloy, i.e., a rolling process and a thermal cycling process. The thicknesses of the $Ti_{50}Ni_{49}W_1$ plates were reduced to 5%, 10%, 20%, and 30% by a cold rolling mill operated at room temperature. In the thermal cycling process, the temperature range and cycling times (N) of the thermal cycle were chosen as 273 to 523 K and $N = 1, 5, 10, 25, 50,$ and 100 cycles. The thermal cycling process comprised cooling the specimens at 273 K for 1 min, resting in air for 1 min, and heating in an air furnace at 523 K for 1 min. After cooling to room temperature in air, the thermal oxide films on the surfaces of the specimens were removed, using #1000 SiC abrasive papers.

The transformation temperatures of the alloys were measured, using a differential scanning calorimeter (DSC, Dupont 2000, METTLER TOLEDO, Zürich, Switzerland), in the range of 173 to 523 K, under heating and cooling rates of 10 K/min. A micro Vickers test was used for measuring the hardness of the polished specimens (10 mm × 1 mm × 1 mm) with a 500 g load. Five readings were recorded, and the average value was obtained. The microstructures of the alloys were observed, using a transmission electron microscope (TEM, PHILIPS-CM200, Hillsboro, OR, USA). The compositions of the alloys were identified by an electron probe microanalyzer (EPMA, JEOL JXA-8900JXA, Akishima, Tokyo). The shape recovery effect was examined by conducting a bending test [23]. Three samples in each group were tested, and the average shape recovery value was estimated. To ensure the alloys had completely transformed into the martensite phase, loading was performed while the specimens were immersed in a liquid nitrogen bath. Specimens (50 mm × 2 mm × 0.5 mm) were bent to a specific angle $\theta_i$ under loading, by a specific mold which has a half-sphere opening. The bent specimens were removed from the mold and then heated to 473 K by a temperature controllable hotplate. After the specimens recovered, $\theta_f$ was measured. The corresponding shape recovery was calculated according to the equation listed below.

$$\varepsilon_s = T/2R$$

$$\text{Shape recovery} = (\theta_i - \theta_f)/\theta_i$$

where $T$ and $R$ are the specimen thickness and the deformed radius, respectively.

## 3. Results and Discussion

### 3.1. Phase Transformation of the Alloys

Figure 1 shows the DSC measurements for the homogenized $Ti_{50}Ni_{49}W_1$ alloy. The martensite peak temperature (*M\**) and the austenite peak temperature (*A\**) were 334 K and 367 K, respectively, and the curve also shows first-order phase transformation of B2 ↔ M.

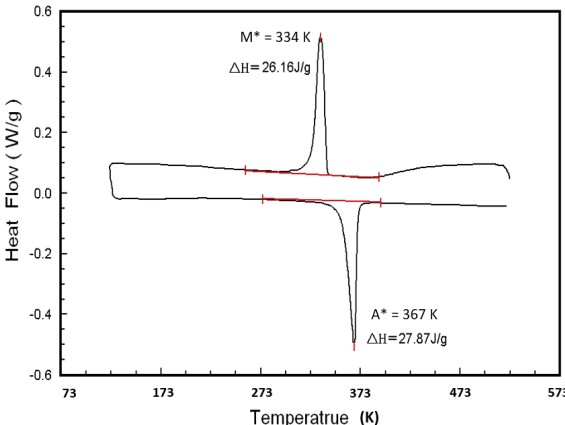

**Figure 1.** DSC curve of as-homogenized $Ti_{50}Ni_{49}W_1$ alloy. *M*\*: The martensite peak temperature. *A*\*: The austenite peak temperature.

Figure 2 shows the backscattering electron image of the as-homogenized $Ti_{50}Ni_{49}W_1$ alloy observed by EPMA. Many black and white particles were distributed in the matrix, and their compositions are listed in Table 1. The matrix, black particles, and white particles are the Ti(Ni,W) phase, $Ti_2$(Ni,W) phase, and tungsten-rich solid solution, respectively, according to the Ti-Ni-W phase diagram at 1400 K [24].

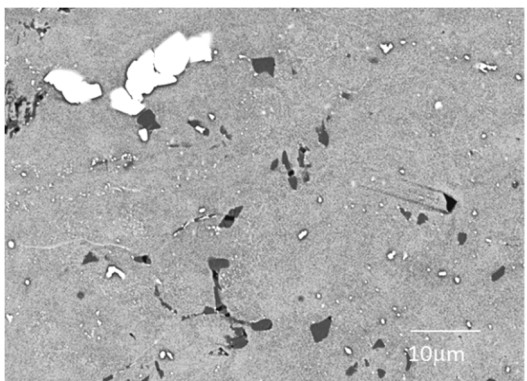

**Figure 2.** EPMA backscattering electron image of the as-homogenized $Ti_{50}Ni_{49}W_1$ alloy.

**Table 1.** Composition of the as-homogenized $Ti_{50}Ni_{49}W_1$ alloy detected from Figure 2.

| Site | Ti (atomic %) | Ni (atomic %) | W (atomic %) | Ti/(Ni + W) | Phase |
|---|---|---|---|---|---|
| Matrix | 49.98 | 49.71 | 0.31 | 0.999 | Ti(Ni,W) |
| Black particle | 66.76 | 32.94 | 0.30 | 2.008 | $Ti_2$(Ni,W) |
| White particle | 15.17 | 3.29 | 81.54 | - | Tungsten-rich solid solution |

The Ti/(Ni+W) ratio in the matrix of the $Ti_{50}Ni_{49}W_1$ alloy is ~1.0 (Table 1), and the peak temperatures of *M*\* and *A*\* are close to those of the $Ti_{50}Ni_{50}$ alloy [5]. However, the second-phase particles in $Ti_{50}Ni_{49}W_1$ alloy are not observed in the $Ti_{50}Ni_{50}$ alloy. It is well-known that the cause of the decrease in the martensite peak temperature (*M*\*) and the martensite start temperature ($M_s$) temperatures is attributed to the oxidation reaction of the TiNi matrix [25]. In the $Ti_{50}Ni_{49}W_1$ alloy, oxygen preferably reacts with the $Ti_2$(Ni,W) phases, and, thus, less oxygen was detected in the matrix. Previous reports have shown [20,26] that $Ti_4Ni_2O$ oxide forms when oxygen reacts with the $Ti_2Ni$ phase. Hence, the precipitation of the second phase is expected to reduce the transformation temperatures. In addition, the martensitic transformation involves a large amount of shear, and any strengthening mechanism can impede the transformation shear, thus lowering the *M*\*($M_s$) temperatures [27,28]. Accordingly, the strengthening due to the presence of W in the TiNi matrix also contributes to

the inhibition of the decrease in transformation temperatures. In conclusion, the abovementioned effects, precipitation of a second phase, and solidification strengthening allow $Ti_{50}Ni_{49}W_1$ to have transformation temperatures similar to $Ti_{50}Ni_{50}$ alloy.

Figure 3a shows that the as-annealed $Ti_{50}Ni_{49}W_1$ alloy exhibited a typical martensite phase. The corresponding selected area diffraction pattern (SADP) of the martensite phase is shown in Figure 3b–d, and the lattice parameters were calculated as $a = 0.301$ nm, $b = 0.423$ nm, $c = 0.472$ nm, and $\beta = 97.5°$, which represents a monoclinic structure. The lattice parameters of the $Ti_{50}Ni_{49}W_1$ alloy are greater than the $Ti_{50}Ni_{50}$ alloy with lattice parameters of $a = 0.2889$ nm, $b = 0.4120$ nm, $c = 0.4622$ nm, and $\beta = 96.8°$ [29]. The DSC and SADP results suggest that the B19' $\leftrightarrow$ B2 transformation sequence occurred in the as-annealed $Ti_{50}Ni_{49}W_1$ alloy.

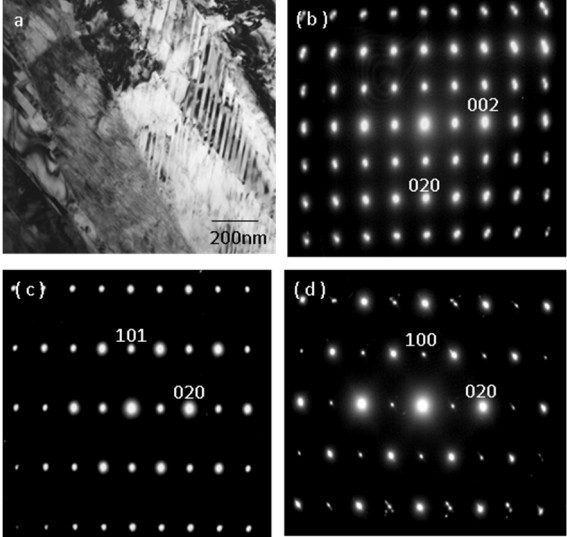

**Figure 3.** (**a**) TEM bright-field image of the as-annealed $Ti_{50}Ni_{49}W_1$ alloy, (**b**) SADP of (a) with $[100]_M$ zone axis, (**c**) SADP of (a) with $[10\bar{1}]_M$ zone axis, and (**d**) SADP of (a) with $[001]_M$ zone axis.

The shape-recovery ability of the $Ti_{50}Ni_{49}W_1$ alloy as a function of temperature is shown in Figure 4. The $Ti_{50}Ni_{49}W_1$ alloy had a slightly lower shape recovery (86%), in comparison with the $Ti_{50}Ni_{50}$ alloy (90%), when the specimens were heated above 493 K [23]. The reason is that the second-phase particles in the $Ti_{50}Ni_{49}W_1$ alloy do not possess shape-memory ability.

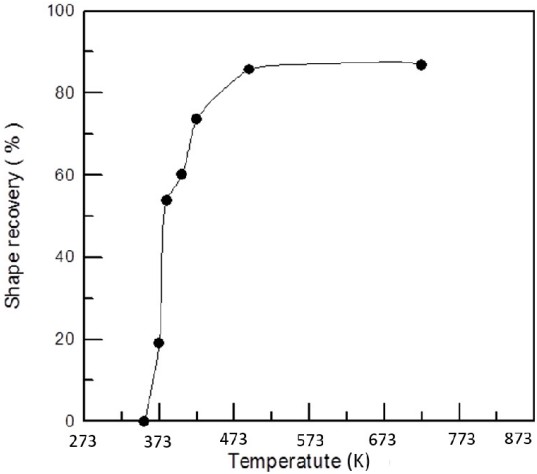

**Figure 4.** Shape recovery of the as-homogenized $Ti_{50}Ni_{49}W_1$ alloy as a function of heating temperature.

### 3.2. The Effects of Thermal Cycling on Phase Transformation

In Figure 5, decreases in the *M\** and *A\** temperatures and an increase in the hardness with increasing thermal cycles were found. This feature stems from the dislocations induced by thermal cycling [30]. The *A\** and *M\** decreased quickly for the first 25 cycles. The decrease of peak temperature of the annealed $Ti_{50}Ni_{49}W_1$ alloy after the first 25 cycles (about 288 K) is greater compared with $Ti_{50}Ni_{50}$ and Ni-rich TiNi binary alloys (≈283 K) [30]. The increase in hardness of this alloy ($\Delta H_v = 31$) is also larger than that of the $Ti_{50}Ni_{50}$ alloy ($\Delta H_v = 24$) at the same number of cycles ($N = 25$) [31]. This difference indicates that rapid dislocation propagation occurred in the $Ti_{50}Ni_{49}W_1$ alloy during the initial stages of thermal cycling. Thermal cycling makes the B19′ ↔ B2 transformation occur in the matrix, but the lattice structure of the second-phase remained unchanged; thus, it induced stress concentrations at the interface. These stress fields promote dislocation propagation and increase the hardness, which decreases the phase-transformation temperatures. In addition, after 25 cycles, the changes in peak temperatures and hardness with the number of cycles is slowed down, which is attributed to the saturation in the amount of dislocations in the $Ti_{50}Ni_{49}W_1$ alloy.

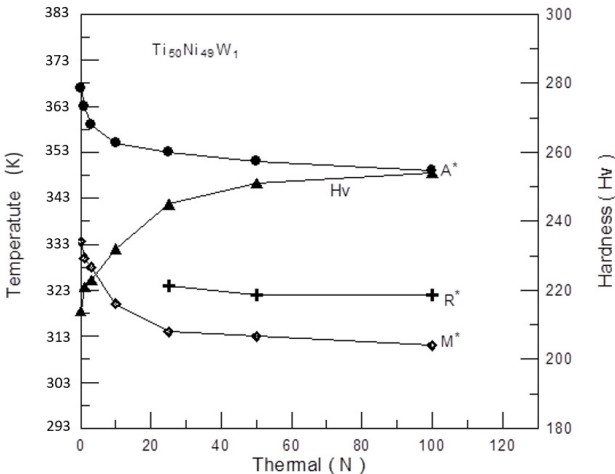

**Figure 5.** Austenite peak temperature (*A\**), martensite peak temperature (*M\**), R-phase peak temperature (*R\**), and hardness of the $Ti_{50}Ni_{49}W_1$ alloy as a function of the number of thermal cycles.

Figure 6 shows that the R-phase transformation can occur during the first 25 cycles. Compared with $Ti_{50}Ni_{50}$ alloy, which needs more than N ≈ 100 cycles for the R-phase transformation to happen [31]. This suggests that the R-phase transformation can occur more easily in the $Ti_{50}Ni_{49}W_1$ alloy than in the $Ti_{50}Ni_{50}$ alloy, during thermal cycling.

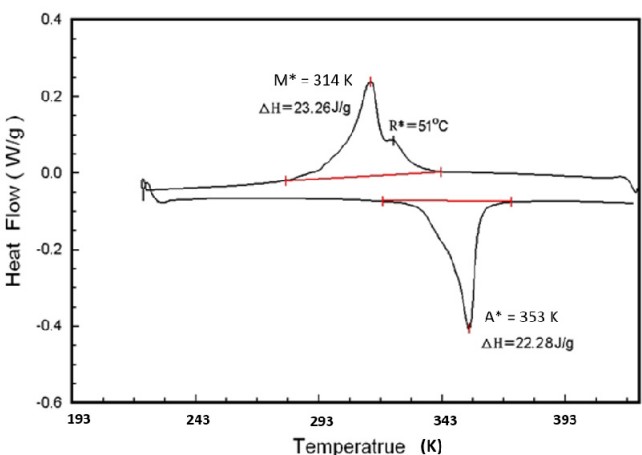

**Figure 6.** DSC curve of the $Ti_{50}Ni_{49}W_1$ alloy after 25 thermal cycles.

### 3.3. The Effects of Cold Rolling on Phase Transformation

In this study, the martensite transformation of the $Ti_{50}Ni_{49}W_1$ alloy undergoing cold rolling at ambient temperature was investigated. Figure 7 shows the DSC curve of the $Ti_{50}Ni_{49}W_1$ alloy after cold rolling to a 5% reduction in thickness, at room temperature. In the first heating cycle (173 K to 503 K), the appearance of the austenite peak temperature in the first heating cycle ($A_1$*) was found at 404 K. In the following cooling period (503 K to ≈373 K), the duplex peak appeared where $M$* is at 318 K and $R$* (B2 → premartensite R phase) is at 327 K. In the second heating (173 K to 503 K) period, the austenite peak temperature in the second heating cycle ($A_2$*) appeared at 358 K, which was significantly lower than the $A_1$* (404 K). The DSC curves for other cold-rolled specimens (10–30%) were similar to those shown in Figure 7 and were therefore not shown here. An increase in the amount of cold rolling caused the peak $A_1$* temperature to significantly increase, but $A_2$* to decreased. The delay of the austenite-phase appearance is considered as mechanically induced martensite stabilization in the $Ti_{50}Ni_{49}W_1$ alloy. This martensite stabilization phenomenon is also in the Ti-rich $Ti_{51}Ni_{49}$ alloy [20]. The martensite stabilization disappeared, and the austenite temperature ($A_2$*) shifted to a lower temperature after the second heating.

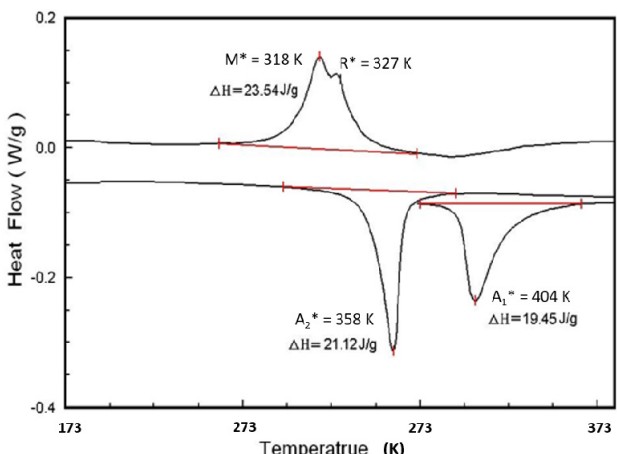

**Figure 7.** DSC curve of the $Ti_{50}Ni_{49}W_1$ alloy after 5% cold rolling.

To analyze the effects of the cold working on the martensite stabilization, the difference between peak temperature $A_1$* and $A_2$*, $\Delta A$*, was shown in Figure 8a. The $Ti_{50}Ni_{49}W_1$ alloy exhibited higher $\Delta A$*, in comparison with the $Ti_{50}Ni_{50}$ alloy [32], for the specimens that underwent 5–30% cold rolling. In addition, the increase in the degree of work-hardening of the $Ti_{50}Ni_{49}W_1$ alloy was higher than that of the $Ti_{50}Ni_{50}$ alloy, as shown in Figure 8b. This is due to the as-annealed $Ti_{50}Ni_{49}W_1$ alloy being harder than $Ti_{50}Ni_{50}$ alloy due to the W atoms in solid solution in the matrix and the presence of the second-phase particles. The second phases in $Ti_{50}Ni_{49}W_1$ alloy can inhibit the deformation of grains. Hence, more load is needed to make $Ti_{50}Ni_{49}W_1$ alloy attain 5–30% cold rolling, which means more dislocations/defects are induced in $Ti_{50}Ni_{49}W_1$ alloy than those in $Ti_{50}Ni_{50}$ alloy during cold rolling. Therefore, the degree of the mechanically induced martensite stabilization in $Ti_{50}Ni_{49}W_1$ alloy is higher than that in $Ti_{50}Ni_{50}$ alloy, as shown in Figure 8.

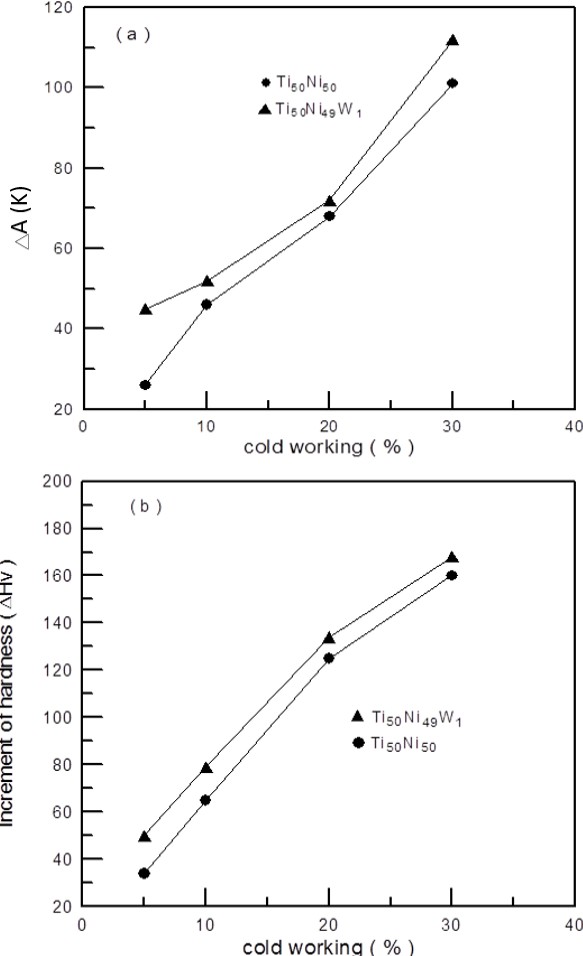

**Figure 8.** Martensite stabilization represented by the difference between the peak temperature $A1^*$ and $A2^*$ ($\Delta A^*$). (**a**) $\Delta A^*$ and (**b**) the increase in hardness of the $Ti_{50}Ni_{49}W_1$ and $Ti_{50}Ni_{50}$ alloy [32] after 5–30% cold rolling.

### 3.4. The Effects of Strengthening on Martensitic Transformation Temperatures

Figure 9a,b shows the peak temperature, $M^*$, as a function of hardness of the thermal-cycled and cold-worked $Ti_{50}Ni_{49}W_1$ and $Ti_{50}Ni_{50}$ alloys, respectively. As mentioned in Section 3.1, the transformation temperatures can be depressed by any strengthening mechanism due to the transformation shear being impeded. This behavior can be represented by Equation (1):

$$M_s = T_o - K\Delta\sigma_y \tag{1}$$

where $T_o$ is the equilibrium temperature, which depends on the composition of the alloy; and K is the constant, which contains the factors of proportionality between the critical shear stress and the yield stress, $\Delta\sigma y$. The yield stress, $\Delta\sigma y$, is proportional to the hardness.

Here, $T_o$ is a constant, since the composition of the alloy does not change under both thermal cycling or cold rolling. In addition, the alloy is strengthened by both thermal cycling and cold rolling, due to dislocation/defect propagation, and an increase in the yield stress is expected. According to Equation (1), strengthening mechanisms, such as thermal cycling and cold rolling, lower the phase transformation temperatures. The slope, K, was calculated according to Equation (1) and marked in Figure 9. Results show that the thermal cycle and cold working induced different K values. In the thermal-cycling process, both the residual stress from volume change and phase transformation shear induce the formation of dislocations. However, dislocations are induced by plastic deformation in

the cold-rolling process. Carefully examining Figure 9a,b under the same strengthening process, we see the *K* value of the $Ti_{50}Ni_{49}W_1$ alloy is higher than that of the $Ti_{50}Ni_{50}$ alloy, since the W-solid solution and the second phase provide the $Ti_{50}Ni_{49}W_1$ alloy with higher hardness. The further works will focus on the effects of thermal cycle on the shape recovery of the alloys combining with microstructure identifications.

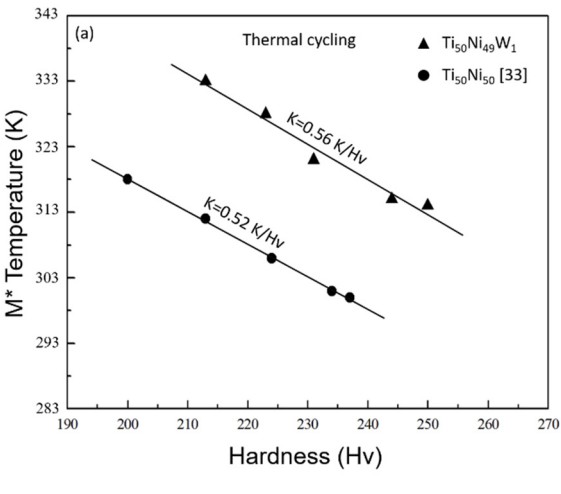

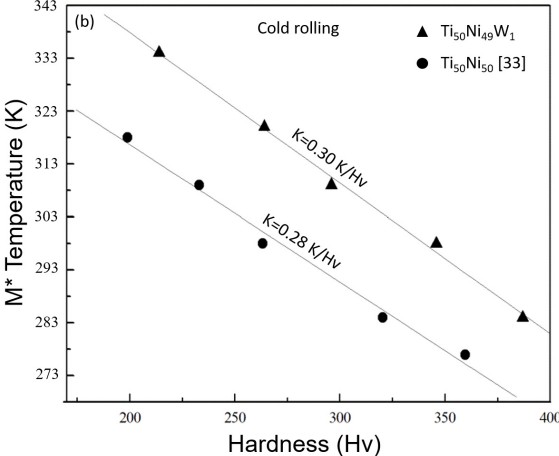

**Figure 9.** *M** temperature versus hardness of $Ti_{50}Ni_{50}$ [33] and $Ti_{50}Ni_{49}W_1$ alloys after (**a**) thermal cycling and (**b**) cold rolling.

## 4. Conclusions

(1) The B19' martensite lattice parameters in the $Ti_{50}Ni_{49}W_1$ alloy are *a* = 0.301 nm, *b* = 0.423 nm, *c* = 0.472 nm, and β = 97.5°. The $Ti_{50}Ni_{49}W_1$ alloy possesses slightly lower shape-recovery ability than the $Ti_{50}Ni_{50}$ alloy due to second-phase particles, $Ti_2(Ni,W)$, precipitating along the grain boundaries of the Ti(Ni,W) matrix.

(2) Peak temperatures of the martensitic and austenite phase were found in lower temperatures and while the hardness values increased with an increasing number of thermal cycles. Furthermore, the decrease in *A** of the $Ti_{50}Ni_{49}W_1$ alloy was larger than that of $Ti_{50}Ni_{50}$ alloy, because the former had a harder matrix and second-phase particles.

(3) Cold rolling causes martensite stabilization of the $Ti_{50}Ni_{49}W_1$ alloy. The increase in the hardness of this alloy is more than that of the $Ti_{50}Ni_{50}$ alloy, which is cold-rolled to the same reduction in thickness. W atoms solid-solvated in TiNi and second-phase $Ti_2(Ni,W)$ are suggested to account for this characteristic behavior.

(4)    Both thermal cycling and cold rolling decrease the *M*s (*M**) temperatures of the $Ti_{50}Ni_{49}W_1$ alloy, and the decrement follows $M_s = T_o - K\Delta\sigma_y$. The *K* value of the $Ti_{50}Ni_{49}W_1$ alloy is greater than that of the $Ti_{50}Ni_{50}$ alloy that underwent the same strengthening process.

**Author Contributions:** Conceptualization, T.-S.H.; methodology, S.-F.O.; software, C.-H.K.; validation, C.-H.Y.; investigation, C.-H.Y.; resources, C.-H.K.; writing—original draft preparation, S.-F.O.; writing—review and editing, C.-H.K.; supervision, C.-H.K. All authors have read and agreed to the published version of the manuscript.

**Funding:** This research received no external funding.

**Conflicts of Interest:** The authors declare no conflict of interest.

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
