# Peer review of "Effects of Thermomechanical Treatment on Phase Transformation of the Ti50Ni49W1 Shape Memory Alloy"

_metals, doi:10.3390/met10040527_

Round 1

Reviewer 1 Report

The overall quality of the research is good as well as the presentation of the results.

Here are some comments:

1) Please change the abstract as T0 and the other parameters in the formula are not defined. You can describe it as a general linear trend.

2) In the introduction, you should improve the state of the art about TiNiW alloys, specifying why it is scientifically interesting to add this element in the NiTi matrix. It is related to the improvement of the mechanical properties/ the functional properties? Please focus and widen these aspects.

2) Please add more info about the bending tests, the setup used and a better explanation of figure 4.

3) About the thermal cycling: is 250 °C the effective temperature reached in the sample (and not only in the furnace)? Did you measure it somehow?

4) Please check the quality of the English language everywere improving the style.

Author Response

Response to Reviewer 1

  • Please change the abstract as T0 and the other parameters in the formula are not defined. You can describe it as a general linear trend.

Reply: The formula was omitted and the sentence was replaced by “The effects of thermal cycling and cold rolling on the Ms temperature of the Ti50Ni49W1 alloy follow a linear trend that Ms decreased with the hardness.”

  • In the introduction, you should improve the state of the art about TiNiW alloys, specifying why it is scientifically interesting to add this element in the NiTi matrix. It is related to the improvement of the mechanical properties/ the functional properties? Please focus and widen these aspects.

Reply: The commonly commercial application of SMAs were shape-memory couplings for aircraft hydraulic connections, such as Cryolive®, Cryofit®, CryoFlare® coupling. TiNiW SMAs were designed for increasing wear resistance of the TiNi SMAs by solidification strengthening.

  • Please add more info about the bending tests, the setup used and a better explanation of figure 4.

Reply: Three samples in each group were tested and the average shape recovery value was estimated. To ensure the alloy had completely changed to martensite phase, the loading was performed on a specimen immersed in a liquid nitrogen bath. Specimens (50mm×2mm×0.5 mm) were bent to a specific angle θi under the loading. After bending, the specimen was removed from the fixture, and then θf was measured, at a temperature above Af. The corresponding shape recovery were calculated according to the equation listen below; here, T and R stand for specimen thickness and deformed diameter, respectively.

εs = T / 2R

Shape recovery = (θi – θf) / θi

  • About the thermal cycling: is 250 °C the effective temperature reached in the sample (and not only in the furnace)? Did you measure it somehow?

Reply: The thermal cycling was done in the chamber of DSC equipment. The sample with weight of 30 mg was put into an aluminum crucibles and the heating rate was controlled as 10 oC/min. The sample is very small with a slow heating rate, and hence, we believe the sample reached to the setting temperature.

      Please check the quality of the English language everywere improving the style.

Reply: The English style and grammar of the manuscript has been revised by native English speakers.

Reviewer 2 Report

The present study investigated the effect of cold work and thermal cycling on the phase transfomation features in W alloyed TiNi. - (L73) You should declare the abbreviation. (A*, M*)
- No precise experimental descriptions on shape memory test (Fig. 4)
- (L107) It is difficult to explain the inferiority of shape memory ability only by the existence of the second phase particles.
  The cofactor conditions which can estimate the reversiblilty in martenstic phase transformation should be considered based on the lattice constants of B2 and B19'.
- (L157)The reviewer can not understand the exact meaning of the sentence("The disloaction~"). Please add corrections.
- Plase add an explanation of how the second phase stabilizes the martensite.
- Plase improve overall English expression (part of speech, tense, word selection, abbreviation format(ex: figure x >> Fig. x) etc)
The reviewer believes that this paper is valuable if the above points are revised.
However, my decision is major revision, now.
Thank you.

Author Response

Response to Reviewer 2

  • The present study investigated the effect of cold work and thermal cycling on the phase transfomation features in W alloyed TiNi. - (L73) You should declare the abbreviation. (A*, M*)

Reply: M*: The martensite peak temperature

A*: The austenite peak temperature

The definition of the abbreviation was added into Fig.1

  • No precise experimental descriptions on shape memory test (Fig. 4)

Reply: Three samples in each group were tested and the average shape recovery value was estimated. To ensure the alloy had completely changed to martensite phase, the loading was performed on a specimen immersed in a liquid nitrogen bath. Specimens were bent to a specific angle θi under the loading. After bending, the specimen was removed from the fixture, and then θf was measured, at a temperature above Af. The corresponding shape recovery were calculated according to the equation listen below; here, T and R stand for specimen thickness and deformed diameter, respectively.

εs = T / 2R

Shape recovery = (θi – θf) / θi

  • (L107) It is difficult to explain the inferiority of shape memory ability only by the existence of the second phase particles.

Reply: Thermal cycling makes B19’«B2 transformation occur in the matrix but the lattice structure of the second-phase maintain unchanged which induces stress concentrations at the interface. This stress fields promotes dislocation propagation and results in an increase in the hardness, which makes phase transformation temperatures decrease.

  • The cofactor conditions which can estimate the reversiblilty in martenstic phase transformation should be considered based on the lattice constants of B2 and B19'.

Reply: Thanks for the suggestions. This study focused on phase transformation characteristics of alloys which are suffered thermal cycling. The maximum thermal cycle was chosen as N = 100 cycles from 0 oC to 250 oC. The range of the applied temperatures were across the temperature of B2 and B19' phase of the TiNiW alloys. The TiNiW alloys under thermal cycles for 100 times still have properties of the reversible phase transformation. Further work will increase the thermal cycle times to make sure the reversiblilty in phase transformation and observe the microstructure of B2 and B19' after thermal cycling.

  • (L157)The reviewer can not understand the exact meaning of the sentence ("The disloaction~"). Please add corrections.

Reply: Thanks for the reminding. The word " disloaction" was corrected by “dislocation”.

  • Plase add an explanation of how the second phase stabilizes the martensite.

Reply: The delay of austenite-phase appearance is considered as the mechanically induced martensite stabilization in Ti50Ni49W1 alloy. Fig8(a) and (b) show the increase in DA* has positive correlation with the increase in harness of the alloys. In addition, the increase in degree of working hardening of the Ti50Ni49W1 alloy is higher than that of Ti50Ni50 alloy, as shown in Fig8(b). This feature results from the as-annealed Ti50Ni49W1 alloy being harder than Ti50Ni50 alloy due to W atoms in solid solution in the matrix, and from the presence of the second phase particles. The second phases in Ti50Ni49W1 alloy can inhibit the deformation of grains. Hence, more load is needed to make Ti50Ni49W1 alloy attain 5-30% cold-rolling, which means more dislocations/defects are induced in Ti50Ni49W1 alloy than those in Ti50Ni50 alloy during cold-rolling. Therefore, the degree of the mechanically induced martensite stabilization in Ti50Ni49W1 alloy is higher than that in Ti50Ni50 alloy.

  • Plase improve overall English expression (part of speech, tense, word selection, abbreviation format(ex: figure x >> Fig. x) etc) The reviewer believes that this paper is valuable if the above points are revised.

Reply: The English style and grammar of the manuscript has been revised by native English speakers.

Reviewer 3 Report

This paper measures the transformation temperatures, hardness and microstructure of the Ti50Ni49W1 alloy with two procedures, rolling process and thermal cycling process, and then compared with Ti50Ni50 alloy. The effects of thermo-mechanical treatment on shape memory effect and phase transformation is evaluated.

Details are required to clarify a more compelling work. Some comments are given as follows.

  1. The title is about the shape memory effect and phase transformation, but few results about shape memory effect in the article are shown and discussed. More discussions should be shown.
  2. Not only Figs. 8 and 9 but also the other figures can be compared with the case of Ti50Ni50 which is a representative shape memory alloy.
  3. The size and the shape of the specimen which is used for hardness test and bending test should be shown.
  4. The measurement method of shape recovery strain including experimental conditions should be mentioned.
  5. In 3.2, I do not find how much shape recovery can be achieved after multiple cycles of thermo-mechanical treatment.
  6. In Fig. 3, more information about (b), (c), (d) should be explained.
  7. In Fig. 4, the temperature range on the abscissa should be up to about 450 degree Celsius. In addition, there is no mention about the temperature range related to shape recovery test pointed out above.
  8. Totally, I could not find any mechanisms why the results can be obtained. The authors must mention the possibility and possible solutions as well as the future works if there are unknown factors.

For minor revisions, the following comments are useful.

  1. The “Fig” in the paragraph should be consistent with the “figure” in the example. For example, the “Fig1” in line 72 is different with the “Figure 1” in line 76 on page #2.
  2. M* and A* in line 127 on page #5 should be specify in the line 73 on page #2 for better understanding. The A1* in line 139, A2* in line 141 on page #5 and M*(Ms) in line 84 on page #3 should be explained as well.
  3. In line 137 on page #5, the subscripts of TiNiW should be marked.
  4. A use of Kelvin for the unit of temperature is strongly recommended.

Author Response

Response to Reviewer3

  • Details are required to clarify a more compelling work. Some comments are given as follows. The title is about the shape memory effect and phase transformation, but few results about shape memory effect in the article are shown and discussed. More discussions should be shown.

Reply: Thanks for the reminding and suggestion about title. We decided to omit “Shape Memory Effect” in Title. The title was replaced by “Effects of Thermomechanical Treatment on Phase Transformation of the Ti50Ni49W1 Shape Memory Alloy”.

  • Not only Figs. 8 and 9 but also the other figures can be compared with the case of Ti50Ni50 which is a representative shape memory alloy.

Reply: This study mainly focused on the effects of thermomechanical treatment on the phase transformation temperatures of Ti50Ni49W1 SMAs. The phase transformation temperatures of the SMAs depend on the strengthening mechanism of the SMAs under thermomechanical treatments. Therefore, the effects of strengthening on martensitic transformation temperatures of Ti50Ni50 SMAs were cited to interpret the results Martensite stabilization of Ti50Ni49W1 SMAs. Therefore, we only compared parts of properties of these two SMAs, such as lattice parameters [A29], hardness, and phase transformation temperatures [33] .

[A29] Otsuka, K. Sawamura, T. Shimizu, K. Crystal structure and internal defects of equiatomic TiNi martensite. Phys. Stat. Sol. 1971, 5 457-470.

  • The size and the shape of the specimen which is used for hardness test and bending test should be shown.

Reply: The size of the specimen for bending and hardness tests is added in the revised manuscript.

bending test: 50 mm × 2 mm × 0.5 mm

Hardness test: 10 mm × 10 mm × 1 mm

  • The measurement method of shape recovery strain including experimental conditions should be mentioned.

Reply: The description of the experimental method of shape recovery is added in the revised manuscript.

Three samples in each group were tested and the average shape recovery value was estimated. To ensure the alloy had completely changed to martensite phase, the loading was performed on a specimen immersed in a liquid nitrogen bath. Specimens (50mm×2mm×0.5 mm) were bent to a specific angle θi under the loading. After bending, the specimen was removed from the fixture, and then θf was measured, at a temperature above Af. The corresponding shape recovery were calculated according to the equation listen below; here, T and R stand for specimen thickness and deformed diameter, respectively.

εs = T / 2R

Shape recovery = (θi – θf) / θi

  • In 3.2, I do not find how much shape recovery can be achieved after multiple cycles of thermo-mechanical treatment.

Reply: The effects of thermal cycles on the shape recovery are the further works. This study focused on the phase transformation temperatures.

  • In Fig. 3, more information about (b), (c), (d) should be explained.

Reply: Fig3(a) shows that the as-annealed Ti50Ni49W1 alloy exhibited a typical martensite phase. The corresponding selected area diffraction pattern (SADP) of the martensite phase is shown in Figs3(b), (c), and (d) and the lattice parameters were calculated as a = 0.301 nm, b = 0.423 nm, c = 0.472 nm, and b = 97.5°, which represents a monoclinic structure. The lattice parameters of the Ti50Ni49W1 alloy are greater than the Ti50Ni50 alloy with lattice parameters of a = 0.2889 nm, b = 0.4120 nm, c = 0.4622 nm, and b = 96.8° [A29]. The description was added into the revised manuscript.

[A29] Otsuka, K. Sawamura, T. Shimizu, K. Crystal structure and internal defects of equiatomic TiNi martensite. Phys. Stat. Sol. 1971, 5 457-470.

  • Totally, I could not find any mechanisms why the results can be obtained. The authors must mention the possibility and possible solutions as well as the future works if there are unknown factors.

Reply: In section 3.4, we used strengthening mechanism of the alloy to interpret the change of phase transformation temperatures when the alloy was undergo cold-cooling. The further works will focus on the effects of thermal cycle on the shape recovery of the alloys. In addition, microstructure identification was applied for understanding the mechanism.

  • For minor revisions, the following comments are useful.

The “Fig” in the paragraph should be consistent with the “figure” in the example. For example, the “Fig1” in line 72 is different with the “Figure 1” in line 76 on page #2.

Reply: Thanks for the reminding. The style of using “Fig” or “Figure” is edited by journal editors.

M* and A* in line 127 on page #5 should be specify in the line 73 on page #2 for better understanding. The A1* in line 139, A2* in line 141 on page #5 and M*(Ms) in line 84 on page #3 should be explained as well.

Reply: The martensite peak temperature (M*) and the austenite peak temperature (A*)

the austenite peak temperature in the first heating cycle (A1*)

the austenite peak temperature in the second heating cycle (A2*)

martensite peak temperature (M*) and the martensite start temperature (Ms)

  • In line 137 on page #5, the subscripts of TiNiW should be marked.

Reply: Thanks for the reminding. The “TiNiW” was replaced by “Ti50Ni49W1”.

  • A use of Kelvin for the unit of temperature is strongly recommended

Reply: Thanks for the suggestions. However, parts of the results of this manuscript are compared with the reference [31,32] which is shown below. Celsius was used in these articles. Therefore, use of Celsius unit is more easily for audiences to read.

[31] Wu, S.K.; Lin, H.C.; Cheng, P.C. Multi-strengthening effects on the martensitic transformation temperatures of TiNi shape memory alloys. J. Mater. Sci., 1999, 34, 5669-3675.

[32] Lin, H.C.; Wu, S.K. Determination of heat of transformation in a cold-rolled martensitic tini alloy. Metall. Trans., 1993, 24A, 293-299.

Round 2

Reviewer 1 Report

The manuscript is now ready for pubblication.

Author Response

Thank you very much!

Reviewer 2 Report

I carefully reviewed the revised manuscript which reflects the my comments last review.
Now, reviewer agree that the present paper is valuable enough to be publised in your jornal.

Author Response

Thank you very much!

Reviewer 3 Report

Thanks for the revisions. I would like to expect the revisions on shape recovery. It is very interested in.

However, the additional descriptions on the following comment are insufficient.

The measurement method of shape recovery strain including experimental conditions should be mentioned.

How to bend the specimen? How to measure θi?

How to measure temperature of the specimen?

What is εs? R should be radius not diameter.

In addition, the following comment is strongly followed.

A use of Kelvin for the unit of temperature is strongly recommended.

This is a world wide standard.

Author Response

  • How to bend the specimen? How to measure θi ?

Reply: The specimen was bent by a specific mold which has a half-sphere opening, as shown below. The bent specimen has a U shape with θi was obtained [23]. The description was added into the revised manuscript.

[23] Lin, H.C.; Wu, S.K., Strengthening effect on shape recovery characteristic of the equiatomic TiNi alloy. Scripta Metall. 1992, 26, 59-62.

2) How to measure temperature of the specimen?

Reply: The bent specimens were heated to 473 K by a temperature controllable hotplate. The specimens recovered to the near original shape when the temperature of the specimen was up to the temperature higher than the Af point. Therefore, we did not measure the recovery temperature here. We measured the recovery temperature (phase transformation temperatures) by DSC.

  • What is εs? R should be radius not diameter.

Reply: R stand for specimen deformed radius. I have corrected that in the revised manuscript.

  • In addition, the following comment is strongly followed. A use of Kelvin for the unit of temperature is strongly recommended. This is a world wide standard.

Reply: The unit of temperature used in this manuscript was corrected to Kelvin. Please see the Figure and content of the revised manuscript. The Figures were shown below.

Round 3

Reviewer 3 Report

Thank you very much for your corrections.